# Determination Position and Initial Value of Aspheric Surface for Fisheye Lens Design

Lirong Fan [1,2,*], Ketao Yan [3,4,*], Guodong Qiao [1], Lijun Lu [2], Shuyuan Gao [3] and Huadong Zheng [2]

[1] School of Aeronautical and Mechanical Engineering, Changzhou Institute of Technology, Changzhou 213032, China; qiaogd@czu.cn
[2] Department of Precision Mechanical Engineering, Shanghai University, Shanghai 200444, China; lulijun@shu.edu.cn (L.L.); bluenote2008@shu.edu.cn (H.Z.)
[3] School of Mechanical Engineering and Rail Transit, Changzhou University, Changzhou 213164, China; 00002839@cczu.edu.cn
[4] The Engineering Research Center for CAD/CAM of Fujian Universities, Putian University, Putian 351100, China
[*] Correspondence: fanlr@czu.cn (L.F.); ketao_yan@163.com (K.Y.)

**Abstract:** The aspheric surface is a commonly used method to improve the imaging quality of the fisheye lens, but it is difficult to determine the position and initial value. Based on the wave aberration theory of the plane-symmetric optical system, a method of using an aspheric surface to design a fisheye lens is proposed, which can quickly determine the appropriate aspheric surface to improve the imaging performance. First, the wave aberration of each optical surface of the fisheye lens is calculated and its aberration characteristics are analyzed. Then, a numerical evaluation function is reported based on the aberration distribution of the fisheye lens on the image plane. According to the functional relationship between the evaluation function and the aspheric coefficient, the position of the aspheric surface and the initial value of the aspheric coefficient can be calculated. Finally, the adaptive and normalized real-coded genetic algorithm is used as the evaluation function to optimize the fisheye lens using an aspheric surface. The proposed method can provide an effective solution for designing a fisheye lens using an aspheric surface.

**Keywords:** fisheye lens design; wave aberration theory; plane-symmetric optical system; aspheric surface; aberrations balance





## 1. Introduction

In fisheye lens system design, spherical surfaces are the most common because of their simple expressions and easier fabrication. But the parameter optimization of a spherical surface only considers the radius of curvature. The aspheric surface needs to determine the radius of the vertex, the quadratic cone coefficient, and other parameters, which makes the aspheric surface have more freedom for parameter optimization than the spherical surface. Thus, using aspheric surfaces can easily correct the optical system aberration and improve the imaging performance [1–6]. However, the processing, inspection, and commissioning of aspheric surfaces are more complicated than spherical surfaces, making the manufacturing cost of optical systems significantly higher [7,8]. To balance the advantages, this paper uses an aspheric surface for a fisheye lens. Reasonable determining position and initial value of the aspheric surface will be significant for improving imaging quality [9].

In the design of the fisheye lens, there are few reports to determine the optimal position and initial value of the aspheric surface. At present, the position of the aspheric surface is usually determined based on experience without theoretical guidance. The fisheye lens usually has a field angle of 180° or even large, and they belong to the plane-symmetric optical systems [10,11]. The Seidel aberration theory cannot be used in aberration analysis. In recent years, Lu et al. have developed an aberration theory of plane-symmetric

optical systems by applying the wave aberration method [10], which can analyze the relationship between structural parameters, including aspheric coefficients, and the imaging performance of a fisheye lens. This theory provides a theoretical basis for determining the position and initial value of the aspheric surface. Based on the theory of plane-symmetric optical systems, we report a method to determine the optimal aspheric position in a fisheye lens and solve for its aspheric coefficient.

The paper is organized as follows: Section 2 introduces the wave aberration theory of plane-symmetric optical systems of an ultrawide-angle optical system. Section 3 describes the calculation method of wave aberration distribution of a single optical plane in a fisheye lens. In Section 4, we define an evaluation function based on the aberrations distribution on the image plane, then determine the sensitive aspheric surfaces and corresponding aspheric coefficients in the fisheye lens system. In Section 5, the method is verified by optimizing a fisheye lens, and compared with the original fisheye lens using the MTF (modulation transfer function) curve. Section 6 summarizes the full text.

## 2. Wave Aberration Theory of the Plane-Symmetric Optical System

### 2.1. The Chief Ray Transfer Equation

The fisheye lens usually has a very large field angle and no longer approximates a paraxial ray. The tracing of a chief ray should use the exact geometrical relationships of trigonometric tracing formulae of a meridian oblique-incident ray. Figure 1 shows a transmission schematic diagram of an off-axis chief ray in the meridian plane. It shows a chief ray $\overline{A_i O_i O_{i+1} O_{i+2}}$ with initial field angle $\omega_{i-1}$ to be refracted by one mirror ($k = i$) and then refracted by two optical surfaces ($k = i + 1$, $k = i + 2$). The chief ray passes through optical media with refractive indices $n_k$, and it impinges optical surface $k$ at $O_k$ and intersects the optical axis at $M_k$. $\omega_k$ means the field angle which is included between the chief ray and the optical axis. The sign of the field angle is identical to the chief ray of the normal in the optical surface. $\alpha_k$ means the angle of incidence, and $\beta_k$ means the angle of reflection or refraction. The sign is identical to the sign of the chief ray to the normal of the optical surface. $\Gamma_k$ and $\rho_k$ are the meridional and sagittal curvature radii of the optical surface $k$ at $O_k$. The distance from the optical surface $k$ to $k + 1$ is $\overline{D_k D_{k+1}} = d_k$. Their signs are positive on the right side of the optical surface. On the left side, they are negative.

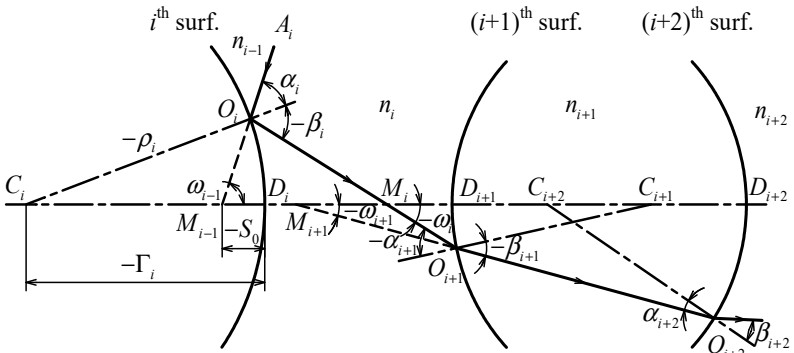

**Figure 1.** The diagram shows a chief ray $\overline{A_i O_i O_{i+1} O_{i+2}}$ passing through a reflecting and two refracting optical surfaces. The chief ray passes through the media of refractive indices $n_{i-1}$, $n_i$, $n_{i+1}$, and $n_{i+2}$ in turn. It impinges optical surface $k$ at $O_k$ and intersects the optical axis at $M_k$. $\omega_k$ means the field angle. $\alpha_k$ and $\beta_k$ mean the angle of incidence and reflection or refraction of optical surface $k$. The optical surfaces are assumed to be quadrics of revolution. The $\Gamma_k$ and $\rho_k$ are the meridional and sagittal curvature radius of the optical surface $k$ at $O_k$. The spacing between the optical surface $k$ and $k + 1$ is $\overline{D_k D_{k+1}} = d_k$.

From Figure 1, the parameters of the chief ray at any optical surface can be calculated with its transfer equation [12]:

$$\sin \alpha_{i+1} = \frac{\Gamma_{i+1} + d_i - \Gamma_i}{\rho_{i+1}} \sin \omega_i + \frac{\rho_i}{\rho_{i+1}} \sin \beta_i, \tag{1}$$

$$\omega_i = \omega_{i-1} + \beta_i - \alpha_i = \omega_0 + \sum_{i=1}^{i} (\beta_i - \alpha_i), \tag{2}$$

$$\beta_{i+1} = \sin^{-1} \left( \frac{n_i}{n_{i+1}} \sin \alpha_{i+1} \right). \tag{3}$$

### 2.2. Transformation of Figure Equation Coordinate System

To apply the plane-symmetric aberration theory [13,14] for calculating the wave aberration of an off-axis point object, we first need to take the coordinate system $xyz$ as the coordinate origin $O$, whose origin $O$ is an intersection of the chief ray and the optical surface, as shown in Figure 2. It shows a chief ray is reflected by a quadric of revolution. The $z$ axis is the normal to the surface and the $x$ axis is along the tangent direction in the meridional plane. $\theta$ is the angle between normal at $O$ and the optical axis.

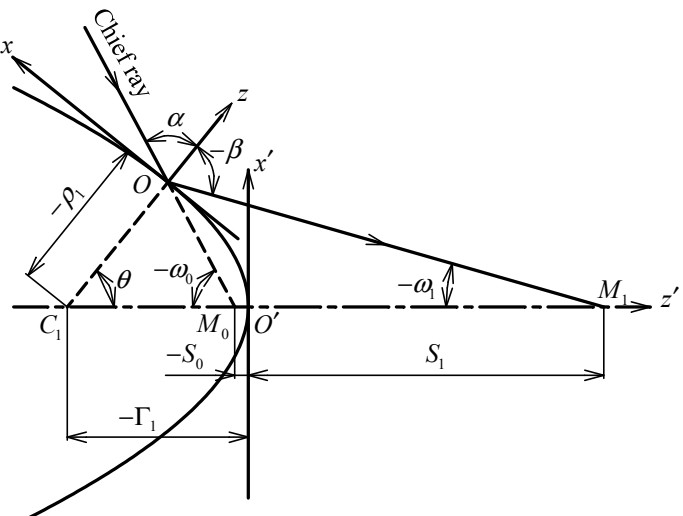

**Figure 2.** The diagram shows a chief ray reflected by a quadric of revolution. The coordinate system of $xyz$. The coordinate system $x'y'z'$ with $O'$ as the origin is transformed into a coordinate system $xyz$, whose origin is at the intersection point of the chief ray to the optical surface $O$. $\theta$ is the angle between the normal at $O$ and the optical axis.

From Figure 2, the expression of the quadrics of revolution in the coordinate system of $x'y'z'$ is expressed as

$$x'^2 + y'^2 = a_1 z' + a_2 z'^2, \ a_1 = 2R_0, \tag{4}$$

where $R_0$ is the curvature radius of the optical surface at point $O'$, $R_0 = \overline{C_1 O'} = -\Gamma_1$. $a_2$ is a constant that determines the type of quadric of revolution. $a_2 > 0$, $a_2 = 0$, $-1 < a_2 < 0$, $a_2 = -1$ and $a_2 < -1$ in Equation (4) represent the quadratic curve as a hyperboloid, a paraboloid, a prolate ellipsoid, a sphere, and an oblate ellipsoid, respectively.

$k$ represents aspheric characteristics in Zemax. The functional relationship between $a_2$ and $k$ is

$$a_2 = -1 - k. \tag{5}$$

The transformation relation from the coordinate systems of $x'y'z'$ to $xyz$ is

$$\left.\begin{array}{l} x' = x\cos\theta + z\sin\theta + x_0^*, \\ y' = y, \\ z' = -x\sin\theta + z\cos\theta + z_0^*. \end{array}\right\} \tag{6}$$

Substituting Equation (6) into Equation (4), we can get the figure equation of the optical surface in the coordinate system of $xyz$; then, its fourth-order Taylor series expansion is

$$z = c_{2,0}x^2 + c_{0,2}y^2 + c_{3,0}x^3 + c_{1,2}xy^2 + c_{4,0}x^4 + c_{0,4}y^4 + c_{2,2}x^2y^2, \tag{7}$$

where $c_{i,j}$ is the figure coefficient [12],

$$
\begin{array}{ll}
c_{2,0} = -\frac{a_1^2}{B^3}, & c_{2,2} = -\frac{2}{B^7}\left(a_1^2 C^2 + 8A^2 x_0^{*2}\right), \\[2mm]
c_{0,2} = -\frac{1}{B}, & c_{4,0} = -\frac{a_1^2\left(a_1^2 C^2 + 16A^2 x_0^{*2}\right)}{B^9}, \\[2mm]
c_{0,4} = -\frac{C^2}{B^5}, & c_{3,0} = -\frac{4Aa_1^2 x_0^*}{B^6}, \\[2mm]
c_{1,2} = -\frac{4Ax_0^*}{B^4}, &
\end{array}
\tag{8}
$$

with

$$
\begin{aligned}
A &= (1 + a_2)\sqrt{4a_2 x_0^* + a_1^2}, \\
B &= \pm\sqrt{4x_0^{*2}(1 + a_2) + a_1^2}, \\
C &= \sqrt{4x_0^{*2} - a_2\left(4a_2 x_0^{*2} + a_1^2\right)}.
\end{aligned}
\tag{9}
$$

From Figure 2, the parameters of $\Gamma_1$ and $\rho_1$ can be derived as

$$
\begin{aligned}
\Gamma_1 &= -\overline{C_1 O'} = z_0^* - \frac{x_0^*}{\tan\theta} = \frac{a_1}{2} + (1 + a_2)z_0^*, \\
\rho_1 &= -\overline{OC_1} = -\frac{\sqrt{4x_0^{*2} + (a_1 + 2a_2 z_0^*)^2}}{2} = \frac{1}{2c_{0,2}}.
\end{aligned}
\tag{10}
$$

The coordinate of $O$ $(x_0^*, y_0^*)$ can be obtained by the geometric relationship of the incident chief ray $\overline{OM_0}$ to the optical surface.

$$
\begin{aligned}
z_0^* &= -\frac{a_1 + 2S_0\tan^2\omega_0 \pm \sqrt{a_1^2 + 4S_0(a_1 + a_2 S_0)\tan^2\omega_0}}{2\left(a_2 - \tan^2\omega_0\right)}, \\
x_0^* &= \tan\omega_0(z_0^* - S_0).
\end{aligned}
\tag{11}
$$

The sign before the root in Equation (11) should be taken as positive for $a_1 < 0$; otherwise, it is negative. The sign of $B$ in Equation (9) should be taken as positive in the case of reflection by a convex mirror or refraction by a concave optical surface, otherwise, it is negative [13].

## 3. Calculation Method of the Wave Aberration of a Fisheye Lens

### 3.1. Aperture Ray Wave Aberrations of an Off-Axis Point Object

For an optical system of $g$ elements, the total wave aberration is [10]

$$W = \sum_{1}^{g}\sum_{ij}^{4} w_{ij0}x^i y^j, \quad (i + j \leq 4, \ j \text{ is even}), \tag{12}$$

with

$$w_{ij0} = nM_{ij0}(\alpha, r_m, r_s, 0) + n'M_{ij0}(\beta, r_m', r_s', 0). \tag{13}$$

In Equation (13), $n$ and $n'$ mean the refraction index of the object and image space. $M_{ij0}$ means the wave aberration coefficient of object beam pencil, which is calculated by Equations (11) and (13)–(16) in ref. [14]. $r'_{m(i)}$ and $r'_{s(i)}$ mean the meridional and sagittal focal distances along the chief ray in the image space of the $g$th optical surface; they can be obtained with the conditions of the wave aberration coefficients, and $w_{200} = 0$ and $w_{020} = 0$ [10] are applied sequentially to every optical surface as

$$
\begin{aligned}
r'_{m(i)} &= \frac{n'_i \cos^2 \beta_i}{2c_{2,0}\left(n_i \cos \alpha_i + n'_i \cos \beta_i\right) - \frac{n_i \cos^2 \alpha_i}{r_{m(i)}}}, \\
r'_{s(i)} &= \frac{n'_i}{2c_{0,2}\left(n_i \cos \alpha_i + n'_i \cos \beta_i\right) - \frac{n_i}{r_{s(i)}}}.
\end{aligned}
\tag{14}
$$

The distance between object points and surface $i + 1$ along the chief ray is given by

$$
r_{m(i+1)} = \overline{d}_i - r'_{m(i)}, \quad r_{s(i+1)} = \overline{d}_i - r'_{s(i)},
\tag{15}
$$

where $\overline{d}_i$ means the optical spacing between the $i$th and $(i + 1)$th optical surfaces along the chief ray.

$$
\overline{d}_i = \frac{\rho_i \sin(\omega_{i-1} - \alpha_i) - \rho_{i+1} \sin(\omega_i - \alpha_{i+1})}{\sin \omega_i}.
\tag{16}
$$

The contributions of the spherical aberration $W_{sph}$ and the coma aberration $W_{coma}$ to any optical surface are

$$
\begin{aligned}
W_{sph} &= w_{400}x^4 + w_{220}x^2y^2 + w_{040}y^4, \\
W_{coma} &= w_{300}x^3 + w_{120}xy^2.
\end{aligned}
\tag{17}
$$

### 3.2. Chief Ray Wave Aberration of an Off-Axis Point Object

The wave aberrations of the field curvature contributed from the first to the $g$th optical surface in the meridional and sagittal direction are [9]

$$
\begin{aligned}
W^*_{m(g)} &= \frac{n'_g \cos^2 \beta_g}{2}\left(\frac{1}{r'_{m(g)}} - \frac{1}{r'_{c(g)}}\right)x^2_g, \\
W^*_{s(g)} &= \frac{n'_g}{2}\left(\frac{1}{r'_{s(g)}} - \frac{1}{r'_{c(g)}}\right)y^2_g.
\end{aligned}
\tag{18}
$$

The wave aberrations of the axial chromatic and transverse chromatic from the first to the $g$th optical surface are [9]

$$
W^*_{CL(g)} = \frac{n'_g}{2}\left(\frac{1}{r'_{F(g)}} - \frac{1}{r'_{C(g)}}\right)x^2_{g\_0},
\tag{19}
$$

$$
W^*_{CT(g)} = \frac{n'_g x_{g\_0}}{r'_{0(g)}}\left(y'_{F(g)} \cos \omega_{F(g)} - y'_{C(g)} \cos \omega_{C(g)}\right).
\tag{20}
$$

In Equation (19), $r'_{F(g)}$ and $r'_{C(g)}$ are the focal distances in the image space of $F$ and $C$ light at normal incidence, and they can be calculated with Equation (14) in the case of $\alpha = 180°$ and $\beta = 0°$. In Equation (20), the height of the chief ray on the image plane for a light of some color is [9,10]

$$
y'_g = \tan \omega_g \left(\frac{\rho_g \sin \beta_g}{\sin \omega_g} - \Gamma_g + r'_{0(g)}\right),
\tag{21}
$$

where $r'_{0(g)}$ represents the distance from the $g$th optical surface to the Gaussian image plane.

The contributions of these types of wave aberrations from the first to the $(g-1)$th optical surface are also calculated with Equations (18)–(20). The wave aberration of field curvature and chromatic contribution from the $g$th optical surface should be calculated by

$$
\begin{aligned}
W_{m(g)} &= W^*_{m(g)} - W^*_{m(g-1)},\\
W_{s(g)} &= W^*_{s(g)} - W^*_{s(g-1)},\\
W_{CL(g)} &= W^*_{CL(g)} - W^*_{CL(g-1)},\\
W_{CT(g)} &= W^*_{CT(g)} - W^*_{CT(g-1)}.
\end{aligned}
\tag{22}
$$

In Equations (17)–(20), $(x_k, y_k)$ is the mapping coordinates of the aperture ray of the beam projected on the $k$th optical surface, and they are obtained by sequential linear approximated from the aperture stop radius [10],

$$
x_k = A_k x_{k+1}, \quad y_k = B_k y_{k+1},
\tag{23}
$$

where,

$$
A_k = \frac{r'_{m(k)} \cos\alpha_{k+1}}{r_{m(k+1)} \cos\beta_k}, \quad B_k = -\frac{r'_{s(k)}}{r_{s(k+1)}}.
\tag{24}
$$

## 4. Determining the Position of Aspheric Surfaces and Aspheric Coefficients

In this section, the optimal aspheric position and initial value of $a_2$ are solved by the functional curve between the evaluation function and the aspheric coefficient.

### 4.1. Evaluation Function

We define an evaluation function that sums types of aberrations at $n$ field angles in its working range:

$$
Q = \sum_{k=1}^{n} \varepsilon_k \left( Q^2_{x(k)} + Q^2_{y(k)} + \eta_k Q^2_{\eta(k)} + \mu_k Q^2_{c(k)} \right),
\tag{25}
$$

and

$$
\begin{aligned}
Q^2_{x(k)} &= \frac{8}{\pi W_q L} \int_{-W_q/2}^{W_q/2} \int_0^{\frac{\sqrt{1-4x^2/W_q^2}}{L/2}} (x' - \overline{x'})^2 dxdy,\\
Q^2_{y(k)} &= \frac{8}{\pi W_q L} \int_{-W_q/2}^{W_q/2} \int_0^{\frac{\sqrt{1-4x^2/W_q^2}}{L/2}} y'^2 dxdy,
\end{aligned}
\tag{26}
$$

with

$$
\overline{x'} = \frac{8}{\pi W_q L} \int_{-W_q/2}^{W_q/2} \int_0^{\frac{\sqrt{1-4x^2/W_q^2}}{L/2}} x' dxdy
\tag{27}
$$

where $\varepsilon_k$ and $\mu_k$ mean the corresponding weight factors of the axial chromatic and transverse chromatic; $Q_{x(k)}$ and $Q_{y(k)}$ are contributed by the aperture aberration of the $k$th field angle on the image plane, which is obtained by integrating and summing the area covered by the beam [13], respectively.

The evaluation function (Equation (25)) is the sum types of aberrations at $n$ field angles in its working field ranges. This includes the effect of aspherical surfaces on the imaging performance of optical systems.

In Equations (26) and (27), the aperture ray aberrations on the image plane G are calculated by [10]:

$$
\begin{aligned}
x' &= \frac{1}{\cos\omega_g} \left( d_{100}x_g + d_{200}x_g^2 + d_{020}y_g^2 + d_{300}x_g^3 + d_{120}x_g y_g^2 \right),\\
y' &= h_{010}y_g + h_{110}x_g y_g + h_{210}x_g^2 y_g + h_{030}y_g^3,
\end{aligned}
\tag{28}
$$

where $x_g$ and $y_g$ mean the ray coordinates on the last optical surface. Since the light beam from an object of an ultrawide-angle impinges on the optical surface with a large incidence angle, the focal line position in the meridian plane and the sagittal plane is severely deviated, and the aperture ray is usually elliptical when transmitted to the last optical surface. Therefore, the integration area in Equations (26) and (27) are the elliptical domain, and $W_q$ and $L$ mean the projection length of the aperture ray on the final optical surface along the meridional ($x$) and sagittal ($y$) directions, respectively.

$Q_{\eta(k)}$ and $Q_{c(k)}$ in Equation (25) mean the numerical estimation of the transverse aberration on the image plane of the axial color aberration and lateral color aberration, respectively.

$$Q_{\eta(k)} = \left| \frac{2x_{g\_0}\left(r'_{F(g)} - r'_{C(g)}\right)}{r'_{0(g)}} \right|, \tag{29}$$

$$Q_{c(k)} = \left| y'_{F(g)} - y'_{C(g)} \right|. \tag{30}$$

In the aberration theory of plane-symmetric optical systems, we calculated aberrations with the parameters of the chief ray parameters $\omega$, $\alpha$, $\beta$, $r_m$, $r_s$, $r'_m$, and $r'_s$ of each optical surface. These parameters are all related to the figure coefficients $c_{i,j}$. According to Equations (8) and (9), $c_{i,j}$ is determined by the parameters $a_1$ and $a_2$ of the optical surface. $a_1 = 2R_0$ represents the radius of curvature of the optical surface at the vertex, and $a_2$ represents the aspheric coefficient of the optical surface. Therefore, the evaluation function $Q$ in Equation (25) is related to the aspheric coefficient $a_2$. We calculate the variation curve of $Q$ with $a_2$ for each optical surface to determine the optimal position of the aspheric surface and the corresponding initial value of $a_2$. This enables the optical system to have the best imaging performance (that is, $Q$ is the minimum).

### 4.2. Wave Aberration Distribution of the Optical Lens

We will apply the aberration theory of plane-symmetric optical systems [9] to calculate the wave aberration distribution of two lenses, lens I and fisheye lens II, as shown in Figures 3 and 4, and Tables 1 and 2 list their optical parameters. The characteristic parameters of the systems are the focal lengths of 33.7 mm and 14.6 mm, the diameters of the aperture stop of 5 mm and 6 mm, and the maximum working field angle of view of 20° and 80°, respectively.

We use Equations (17)–(22) to calculate the wave aberration distribution of each optical surface of lens I at $\omega_0 = 20°$ and fisheye lens II at $\omega_0 = 80°$. Figures 5 and 6 show $W_m$, $W_s$, $W_{CL}$, $W_{CT}$, $W_{sph}$, and $W_{coma}$, respectively.

From Figures 5 and 6, it can be seen that the total wave aberrations of lens I and fisheye lens II are more serious. The effects of field curvature, lateral color aberration, and coma aberration are particularly significant. The position of the fifth surface is the aperture stop, and its aberration contribution is zero.

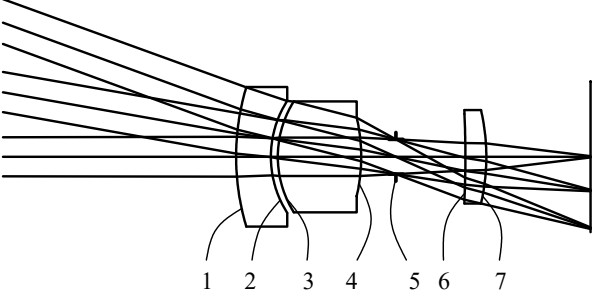

**Figure 3.** The optical system of lens I is an objective lens. Table 1 lists its optical parameters. The numbers in the figure represent the optical surfaces, where the fifth surface is the aperture stop.

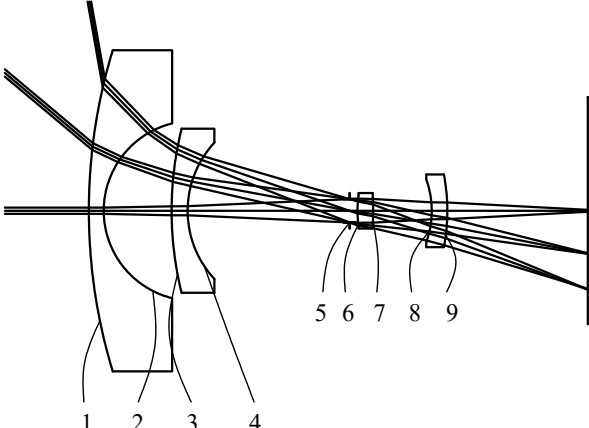

**Figure 4.** The optical system of lens II is a fisheye lens. Table 2 lists its optical parameters. The numbers in the figure represent the optical surfaces, where the fifth surface is the aperture stop.

**Table 1.** Optical parameters of lens I.

| Surface $i$ | Radius/mm | Spacing/mm | Index | Glass |
|---|---|---|---|---|
| Object | Infinite | 2000.0 | | |
| 1 | 35.0 | 5.0 | 1.84666 | N-SF57 |
| 2 | 15.0 | 1.0 | | |
| 3 | 15.0 | 12.0 | 1.45600 | N-FK58 |
| 4 | −25.0 | 5.0 | | |
| 5 (STO) | Infinite | 10.0 | | |
| 6 | −130.0 | 3.0 | 1.68893 | P-SF8 |
| 7 | −30.0 | 15.0 | | |

**Table 2.** Optical parameters of fisheye lens II.

| Surface $i$ | Radius/mm | Spacing/mm | Index | Glass |
|---|---|---|---|---|
| Object | Infinite | 2000.0 | | |
| 1 | 141.445 | 3.808 | 1.71300 | N-LAK8 |
| 2 | 23.059 | 17.392 | | |
| 3 | 87.889 | 4.099 | 1.71300 | N-LAK8 |
| 4 | 25.685 | 41.513 | | |
| 5 (STO) | Infinite | 2.0 | | |
| 6 | 30.0 | 4.0 | 1.71300 | N-LAK8 |
| 7 | −80.0 | 15.0 | | |
| 8 | −24.0 | 4.0 | 1.45600 | N-FK58 |
| 9 | −50.0 | 31.101 | - | - |

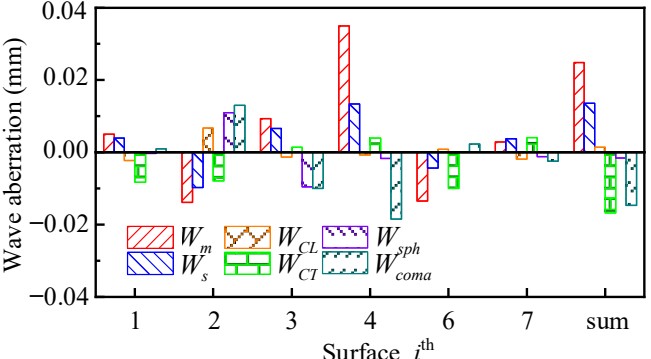

**Figure 5.** The diagram shows the wave aberration distribution of lens I at $\omega_0 = 20°$. Table 1 lists its optical parameters.

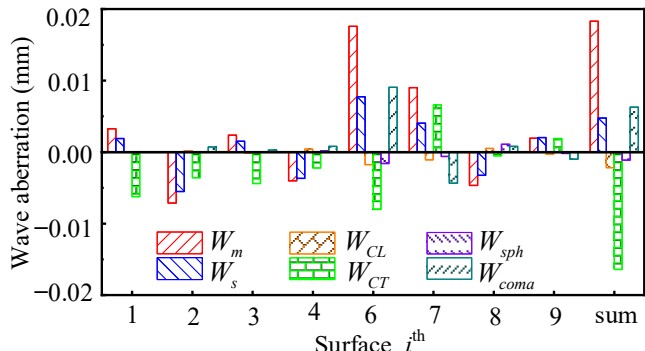

**Figure 6.** The diagram shows the wave aberration distribution of fisheye lens II at $\omega_0 = 80°$. Table 2 lists its optical parameters.

We use Equation (25) to calculate the curve of the $Q$ with the aspheric coefficient $a_2$ of each optical surface of lens I, as shown in Figure 7. Since $a_2$ in the range of [–3, 1] can cover common easily machinable aspheric surface types [15]. The search range of $a_2$ is $[-3, 2]$ with a step of 0.01, and the field angles in Equation (25) are set to $0°$, $5°$, $10°$, $15°$, and $20°$. Weighting factors $\varepsilon_k$, $\eta_k$, and $\mu_k$ are set to 1.

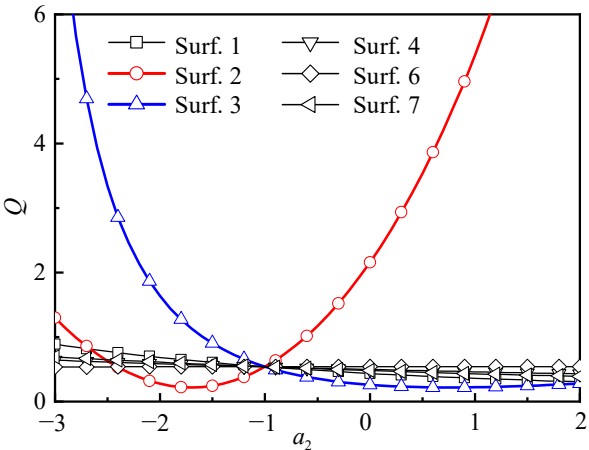

**Figure 7.** The abscissa indicates the aspheric coefficient $a_2$; the ordinate indicates the evaluation function $Q$. The curve shows the relationship between $Q$ and $a_2$ of lens I.

Similarly, the curve of the $Q$ with $a_2$ of each optical surface of fisheye lens II is shown in Figure 8. The search range of $a_2$ is $[-4, 2]$ with a step of 0.01, and the field angles are set to $0°$, $10°$, $20°$, $30°$, $40°$, $50°$, $60°$, $70°$, and $80°$. $\varepsilon_k$, $\eta_k$, and $\mu_k$ are set to 1.

From Figure 7, the second and third optical surfaces of lens I are more sensitive to their imaging performance (making $Q$ reach a minimum). From Figure 8, the second and fourth optical surfaces of fisheye lens II are more sensitive to their imaging performance (making $Q$ reach a minimum), respectively. The corresponding initial value of $a_2$ is shown in Table 3.

**Table 3.** Optimal optical aspheric position in lens I and fisheye lens II, and the corresponding initial value of $a_2$.

|  | Lens I |  | Fisheye Lens II |  |
| --- | --- | --- | --- | --- |
| Surface | 2 | 3 | 2 | 4 |
| $a_2$ | −1.69 | 0.79 | −2.12 | −3.42 |

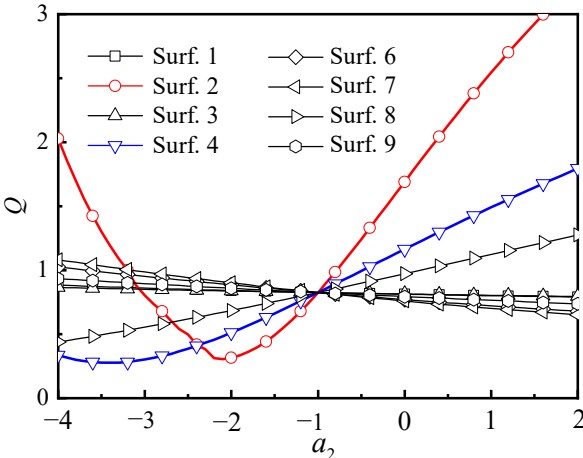

**Figure 8.** The abscissa indicates the aspheric coefficient $a_2$; the ordinate indicates the evaluation function $Q$. The curve shows the relationship between $Q$ and $a_2$ of fisheye lens II.

The wave aberration distributions of lens I and fisheye lens II in the case of $\omega_0 = 20°$ and $\omega_0 = 80°$ are recalculated with Equations (17)–(22), respectively. As shown in Figures 9 and 10, only the $a_2$ of the sensitive optical surface is changed according to Table 3. Figure 9a,b show lens I with $a_{2(2)} = -1.69$ and $a_{2(3)} = 0.79$. Figure 10a,b show fisheye lens II with $a_{2(2)} = -2.12$ and $a_{2(4)} = -3.42$. Compared with Figures 5 and 6, their total wave aberrations decreased significantly, respectively.

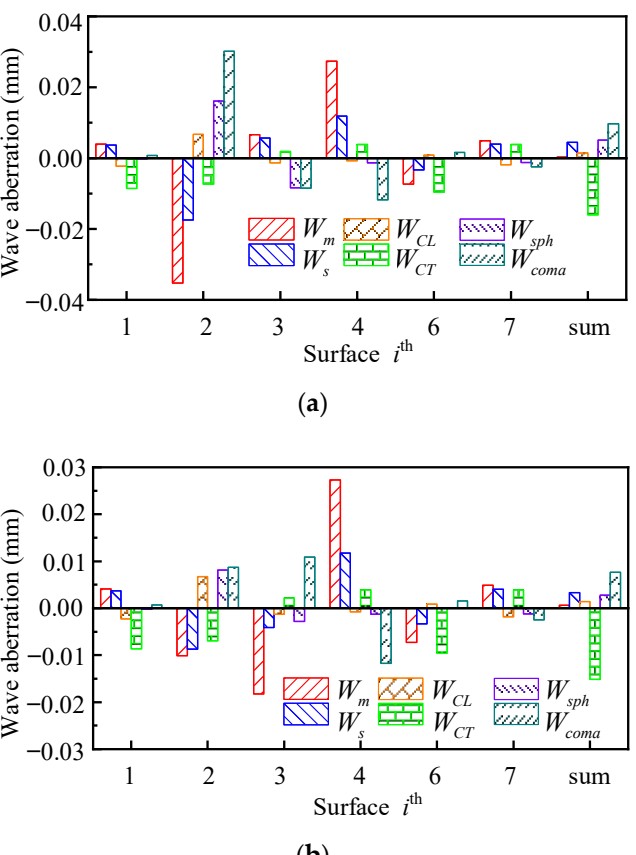

**Figure 9.** The diagram shows the wave aberration distributions of lens I at $\omega_0 = 20°$. They are recalculated with Equations (17)–(22), and only the $a_2$ of the sensitive optical surface is changed. (**a**) shows $a_{2(2)} = -1.69$ and (**b**) shows $a_{2(3)} = 0.79$, respectively.

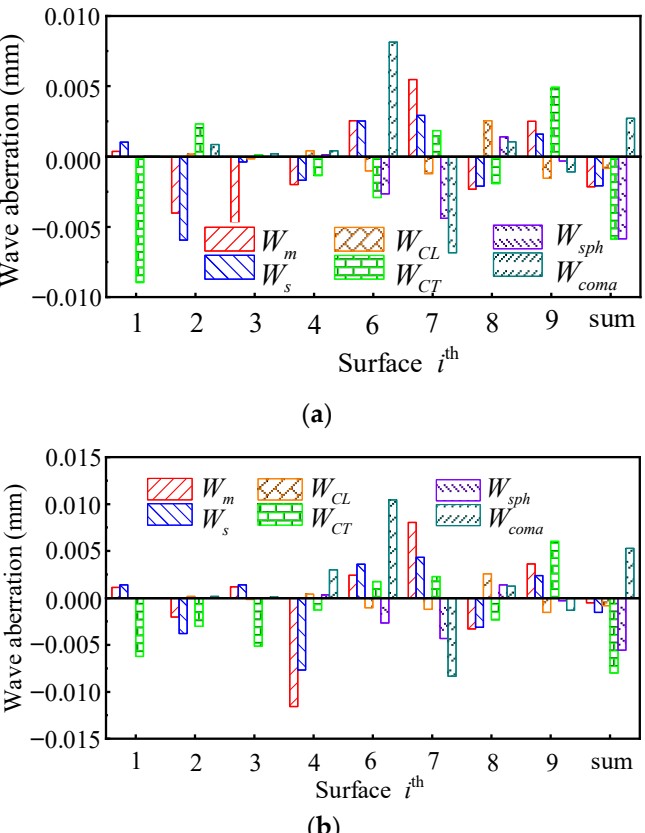

**Figure 10.** The diagram shows the wave aberration distributions of fisheye lens II at $\omega_0 = 80°$. (**a**) shows $a_{2(2)} = -2.12$ and (**b**) shows $a_{2(4)} = -3.42$, respectively.

Compared to Figures 9 and 10 with Figures 5 and 6, in lens I and fisheye lens II, only the aspheric coefficient of one sensitive surface is modified, which makes the total wave aberration of these two lenses significantly reduced. It is confirmed that the use of an aspheric surface at a sensitive optical surface has a significant effect on correcting lens aberrations.

Lens I and fisheye lens II have not been designed and optimized. They only contain three and four lenses, respectively, so their aberrations are more serious. Lens I and fisheye lens II are difficult to optimize the lens system with better performance. We will optimize the fisheye lens using an aspheric surface in the next section.

## 5. Numerical Validation

To verify the performance, we optimize fisheye lens III using an aspheric surface in the following steps: the sensitive optical surface in fisheye lens III and the corresponding initial value of $a_2$ are solved; then, based on the MTF calculated by the aberration theory as the evaluation function, a self-adaptive and normalized real-coded genetic algorithm is used to optimize the optical parameters of fisheye lens III. Note that the optimization parameters do not include the refractive index of the lens's material.

### 5.1. Determining the Position and Initial Value of Aspheric Surface in the Fisheye Lens

Fisheye lens III [16] is composed of 18 standard optical spheres, as shown in Figure 11, where the 11th surface is the aperture stop. Table 4 lists optical parameters, with a focal length of 9 mm, F/# = 5, $2\omega_0 = 160°$.

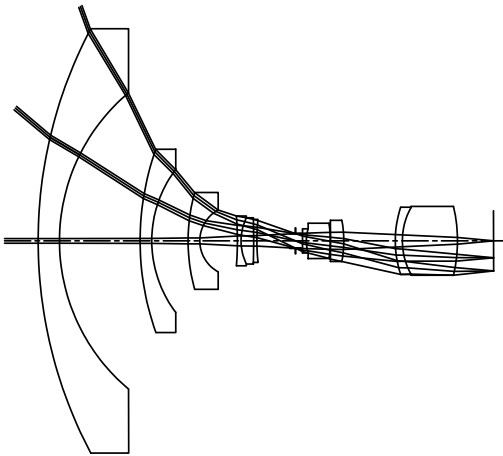

**Figure 11.** The diagram shows the layout of fisheye lens III. Table 4 lists its optical parameters.

**Table 4.** Optical parameters of the original and optimized fisheye lens III.

| Surface $i$ | Radius/mm | | Spacing/mm | | Index | Glass |
|---|---|---|---|---|---|---|
| | Original | Optimized | Original | Optimized | | |
| 1 | 162.236 | 148.852 | 7.676 | 5.946 | 1.5163 | BK7HT |
| 2 | 69.168 | 35.582 | 28.975 | 29.988 | - | - |
| 3 | 98.314 | 60.443 | 4.186 | 3.716 | 1.6127 | SK4 |
| 4 | 42.474 | 35.560 | 13.054 | 10.810 | - | - |
| 5 | 64.484 | 41.520 | 4.210 | 3.521 | 1.6148 | SSK3 |
| 6 | 12.623 | 13.644 | 13.536 | 13.741 | - | - |
| 7 | −105.027 | −170.440 | 1.297 | 1.219 | 1.4875 | N-FK5 |
| 8 | 18.953 | 14.957 | 4.535 | 5.050 | 1.7847 | SF56A |
| 9 | −232.713 | −273.525 | 1.175 | 1.118 | 1.7555 | P-LAF37 |
| 10 | 75.791 | 74.989 | 13.862 | 12.265 | - | - |
| 11 (STO) | ∞ | ∞ | 2.528 | 2.843 | - | - |
| 12 | 57511.3 | 53039.16 | 1.364 | 1.396 | 1.7847 | SF56A |
| 13 | 15.425 | 13.347 | 8.586 | 7.203 | 1.7433 | N-LAF35 |
| 14 | −46.059 | −63.238 | 0.142 | 0.117 | - | - |
| 15 | −259.655 | −316.890 | 4.751 | 4.043 | 1.7555 | P-LAF37 |
| 16 | −37.371 | −36.438 | 18.493 | 16.484 | - | - |
| 17 | 38.3731 | 34.778 | 2.524 | 2.169 | 1.7847 | SF56A |
| 18 | 24.438 | 19.829 | 19.820 | 23.887 | 1.6204 | N-SK16 |
| 19 | −53.609 | −43.558 | 12.963 | 12.006 | - | - |

Taking the aspheric coefficient $a_2$ of each optical surface as a variable in sequence, the evaluation function $Q$ of fisheye lens III is calculated by Equation (25). The minimum value of each curve and the corresponding initial value of $a_2$ are calculated, and the curve is shown in Figure 12. We set the field angles in Equation (25) to be 0°, 10°, 20°, 30°, 40°, 50°, 60°, 70°, and 80°. The weighting factors of $\varepsilon_k$, $\eta_k$, and $\mu_k$ are set to 1.

In Figure 12, the sixth optical surface using an aspheric surface is most sensitive to the imaging performance of fisheye lens III. It is a more significant effect than the other 17 optical surfaces. Therefore, we determine the sixth optical surface as the best aspheric surface position in fisheye lens III, where $a_{2(6)}$ is 1.04 and $Q_{(6)}$ is 0.0385.

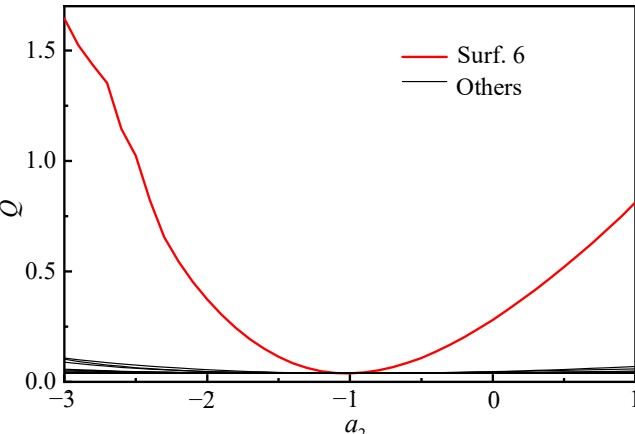

**Figure 12.** The abscissa indicates the aspheric coefficient $a_2$; the ordinate indicates the evaluation function $Q$. The curve shows the relationship between $Q$ and $a_2$ of fisheye lens III.

### 5.2. Optimizing Fisheye Lens III Using an Aspheric Surface

Based on the fourth-order wave aberration theory of the plane-symmetric optical systems, we use the MTF as the evaluation function and apply a self-adaptive and normalized real-coded genetic algorithm to optimize the optical parameters of fisheye lens III. The evaluation function is defined as [12]

$$Q = \eta Q_m + \xi Q_s, \tag{31}$$

and

$$Q_m = \sum_1^k \varepsilon_i \text{MTF}_{m(i)}, \quad Q_s = \sum_1^k \varepsilon_i \text{MTF}_{s(i)}, \tag{32}$$

where $Q_m$ and $Q_s$ mean the evaluation function; and $\text{MTF}_{m(i)}$ and $\text{MTF}_{s(i)}$ mean modulation transfer function in the meridional and sagittal directions, respectively. $\varepsilon_i$ is the corresponding weight factor at the $i$th field angle. In this optimization, the field angle and the corresponding $\varepsilon_i$ are listed in Table 5. We set the weighting factors of the meridional and sagittal directions as $\eta = 4$, $\xi = 1$, respectively.

**Table 5.** Weight factors at different field angles of fisheye lens III.

| | $\omega_i(°)$ | 0 | 10 | 20 | 30 | 40 | 50 | 60 | 70 | 80 |
|---|---|---|---|---|---|---|---|---|---|---|
| Frist round | $\varepsilon_i$ | 1 | 1 | 1 | 1 | 2 | 2 | 2 | 2 | 2 |
| Second round | $\varepsilon_i$ | 1 | 1 | 2 | 2 | 4 | 4 | 6 | 6 | 7 |

The optimization of fisheye lens III is performed in two rounds. In the first round, we search for the optimal values of the upper limit is set to 1.7 times, and the lower limit is set to 0.5 times the original radius of fisheye lens III. The upper limit is set to 1.3 times and the lower limit is set to 0.7 times the original optical distance of fisheye lens III; the search range for $a_2$ is set to $[-2, 0]$. The optimal position of the image plane is set to a range of $\pm 0.5$ mm around $r'_0$.

The number of generations in the genetic algorithm is taken to be 600. We run 20 times for the original fisheye lens III and choose two designs of good performance as the starting designs for the second round of optimization.

In the second round of optimization, we take a relatively small search range of parameters, setting the radius to $\pm 10\%$ and the optical distances to $\pm 5\%$ of the starting designs. Around $\pm 0.2$ of the starting design, the search range of $a_2$ is set. The optimal position of the image plane is in a small range of $\pm 0.2$ mm around $r'_0$. The number of generations in the genetic algorithm is taken to be 200, and each starting design is run 10 times. The optimization result for the aspheric coefficient is $a_{2(6)} = -1.065$. Consequently, we obtained the best

design of the fisheye lens from the optimization, and Table 4 lists its optical parameters. We apply the FFT method of Zemax [17,18] to calculate the MTF curves of the original and optimized fisheye lens III, respectively. We apply Zemax to calculate its MTF curves for the spatial frequencies of 10 lp/mm and 30 lp/mm in the case of F/# = 5 as shown in Figure 13; (a) shows the MTF curve of the original fisheye lens III; (b) shows the MTF curve of the optimized fisheye lens III. By comparing the MTF curves, the proposed approach in the paper makes the imaging performance of fisheye lenses superior. Our next step will be actual production and further testing and research.

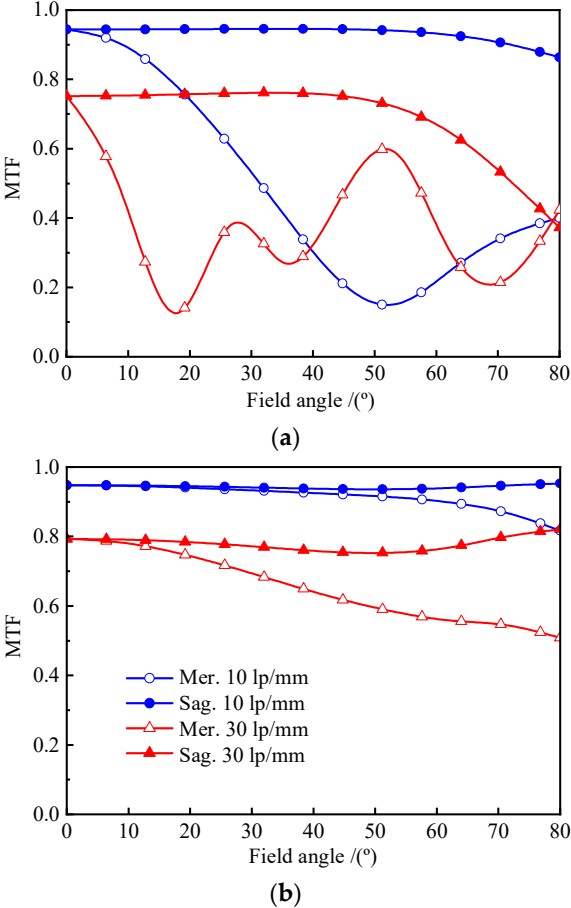

**Figure 13.** The diagram shows the MTF curves of fisheye lens III. (**a**) shows the MTF curve of the original fisheye lens III; (**b**) shows the MTF curve of the optimized fisheye lens III, with $a_{2(6)} = -1.065$. Their optical parameters are listed in Table 5.

From Table 4, compared with the original optical parameters of fisheye lens III, the optimized parameters have little change. The aspheric coefficient $a_{2(6)} = -1.065$ of the sixth optical surface is close to the standard spherical surface. From Figure 13, we obtain the best design of fisheye lens III from the optimization.

## 6. Conclusions

Based on the wave aberration theory of plane-symmetric optical systems, this paper develops a method by which to determine the position and initial value of the aspheric surface, which provides an effective means for the design and optimization of fisheye lens systems. Through aberration calculation and example verification, the conclusions are as follows:

(1)    In fisheye lenses, the optical surface with a small curvature radius generally adopts an aspheric surface, which is more advantageous for improving its imaging performance.

The aspheric coefficient is also closer to the spherical surface, making it easy to manufacture.

(2) In the case of a moderate acceptance aperture of fisheye lenses, the contribution of the aspheric surface to the balance of field curvature aberration is more significant.

The method in this paper is also suitable for the optimal design of paraxial optical imaging systems and other wide angle lens systems. We hope that this work may be helpful to optical designers in creating high-quality fisheye lens systems.

**Author Contributions:** Conceptualization, L.F. and L.L.; methodology, L.F.; software, L.F. and K.Y.; validation, G.Q. and L.F.; formal analysis, S.G. and H.Z.; investigation, L.F.; resources, L.F.; data curation, K.Y.; writing—original draft preparation, L.F.; writing—review and editing, L.F.; visualization, G.Q.; supervision, L.L. and K.Y.; project administration, K.Y.; funding acquisition, K.Y. All authors have read and agreed to the published version of the manuscript.

**Funding:** National Natural Science Foundation of China (NSFC) (61975111); National Natural Science Foundation of China (NSFC) (52205552); Open Project of The Engineering Research Center for CAD/CAM of Fujian Universities (K202206); Basic Research Program of Jiangsu Province (Natural Science Foundation)—Youth Fund Project (BK20200983).

**Institutional Review Board Statement:** Not applicable.

**Informed Consent Statement:** Not applicable.

**Data Availability Statement:** Data underlying the results presented in this paper are not publicly available at this time but may be obtained from the authors upon reasonable request.

**Conflicts of Interest:** The author declares that there is no conflict of interest related to this article.

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
