# Peer review of "Determination Position and Initial Value of Aspheric Surface for Fisheye Lens Design"

_photonics, doi:10.3390/photonics10121381_

Round 1
Reviewer 1 Report
Comments and Suggestions for Authors
There are some questions for that paper.
1. In this paper, what’s the reasons for using optical systems 1 and 2 for algorithm validation and optical system 3 for optimization of the design?
2. Can authors give the basis for the selection of certain data in the text, e.g. number of generations in the genetic algorithm, and the amount of variation of the radius, and the optical distance appearing in section 5.2.
3. Optical design software such as zemax and codev already have the ability to optimize optical systems by calculating the wave aberration for each plane in the optical system. What is the improvement or advantage of the algorithms mentioned in the paper over the algorithms in optical design software?
Reviewer 2 Report
Comments and Suggestions for Authors
This manuscript proposes to use wave aberration theory for plane-symmetric optical system to derive the wave aberration fisheye lens system, so that which optical system should be assigned as an aspherical surface and the aspheric coefficient can be determined based on the sensitivity to the wavefront error. It can be considered as one approach for designing fisheye lens system, but some argument and fundamental issues need to be addressed to make the proposed concept more convincing.
1. The way for deriving wave aberration is using exact ray tracing to find the optical path difference between the chief ray and the skew ray. There is no clear description to show which field point has been used, on axis field point or off-axis field point? If off-axis field point, which off-axis field point is used. Fig. 1 and Fig. 2 are not quite consistent.
2. When doing sensitivity analysis, the sensitivity to wavefront aberration can be different for different field point. Is it adequate to use the sensitivity analysis for single field point to determine which surface to be aspheric and the associated aspheric coefficient? It can be even less adequate for the surface where several ray cones from different field point covering the same surface area.
3. When using the proposed approach for designing a fisheye lens, a well layout of the whole optical system need to be ready. At this stage, one can easily set the conic constant as a variable, and make optimization with commercial ray tracing program to see whether the image quality for all field points can be significantly improved. Trying one optical surface just takes a few seconds. If with experience, one could even determine directly which surface could be more effective as an aspheric based on which kind aberration is included and the Seidal coefficient of each surface. Can the design efficiency be improved with the proposed approach?
4. There have been quite a few fisheye lens system successfully used in practice. Is it possible to provide evidence to show that using the proposed approach can lead to a better fisheye lens design with a clear definition of performance index.
5. Is the proposed concept applicable to other wide angle lens system? not just fisheye lens system.
Comments on the Quality of English Language
N/A
Round 2
Reviewer 1 Report
Comments and Suggestions for Authors
the revised paper have met the requirements for acceptance
Author Response
Dear Reviewer,
Thank you very much for your efforts in handling our manuscript.
Reviewer 2 Report
Comments and Suggestions for Authors
The authors have addressed the issues raised in the previous review, and the manuscript can be considered for the publication in Photonics. However, it is suggested that the statement given in the authors' response should be put into the manuscript, so that the readers can know how the aberration data is generated. For example, how many and which field points have been chosen for ray tracing to calculate the aberration coefficient and sensitivity.
Comments on the Quality of English LanguageN/A
Author Response
Dear Reviewer,
Thank you very much for your efforts in handling our manuscript. According to your comments, the corresponding changes have been highlighted in red in the manuscript.